# Patterns of Healthcare Resource Utilisation of Critical Care Survivors between 2006 and 2017 in Wales: A Population-Based Study

**DOI:** 10.3390/jcm12030872

**Published:** 2023-01-21

**Authors:** Mohammad Alsallakh, Laura Tan, Richard Pugh, Ashley Akbari, Rowena Bailey, Rowena Griffiths, Ronan A. Lyons, Tamas Szakmany

**Affiliations:** 1Population Data Science, Swansea University Medical School, Faculty of Medicine, Health and Life Science, Swansea University, Swansea SA2 8PP, UK; 2Department of Anaesthesia, Intensive Care and Pain Medicine, Division of Population Medicine, Cardiff University, Cardiff CF14 4XN, UK; 3Department of Anaesthetics, Glan Clwyd Hospital, Betsi Cadwaladr University Health Board, Rhyl LL18 5UJ, UK; 4Critical Care Directorate, Grange University Hospital, Aneurin Bevan University Health Board, Cwmbran NP44 8YN, UK

**Keywords:** critical care survivorship, healthcare resource utilisation, Wales

## Abstract

In this retrospective cohort study, we used the Secure Anonymised Information Linkage (SAIL) Databank to characterise and identify predictors of the one-year post-discharge healthcare resource utilisation (HRU) of adults who were admitted to critical care units in Wales between 1 April 2006 and 31 December 2017. We modelled one-year post-critical-care HRU using negative binomial models and used linear models for the difference from one-year pre-critical-care HRU. We estimated the association between critical illness and post-hospitalisation HRU using multilevel negative binomial models among people hospitalised in 2015. We studied 55,151 patients. Post-critical-care HRU was 11–87% greater than pre-critical-care levels, whereas emergency department (ED) attendances decreased by 30%. Age ≥50 years was generally associated with greater post-critical-care HRU; those over 80 had three times longer hospital readmissions than those younger than 50 (incidence rate ratio (IRR): 2.96, 95% CI: 2.84, 3.09). However, ED attendances were higher in those younger than 50. High comorbidity was associated with 22–62% greater post-critical-care HRU than no or low comorbidity. The most socioeconomically deprived quintile was associated with 24% more ED attendances (IRR: 1.24 [1.16, 1.32]) and 13% longer hospital stays (IRR: 1.13 [1.09, 1.17]) than the least deprived quintile. Critical care survivors had greater 1-year post-discharge HRU than non-critical inpatients, including 68% longer hospital stays (IRR: 1.68 [1.63, 1.74]). Critical care survivors, particularly those with older ages, high comorbidity, and socioeconomic deprivation, used significantly more primary and secondary care resources after discharge compared with their baseline and non-critical inpatients. Interventions are needed to ensure that key subgroups are identified and adequately supported.

## 1. Introduction

With advances in individualised care and the adoption of evidence-based practice, more patients are surviving critical illness. However, survivors of critical illness are known to sustain higher levels of physical, functional, and cognitive disability and impairment, with accompanying ongoing complex healthcare needs [1,2,3,4]. As a result, they often suffer from a significant burden of morbidity following discharge; indeed, some impairments persist for many years, and mortality remains higher than a comparative non-critical hospitalised population [5,6,7,8]. The COVID-19 pandemic has accelerated these trends, with significant numbers of individuals surviving acute respiratory distress syndrome (ARDS) and critical illness. As such, critical care survivorship appears set to become a major population health concern, with increasing demand on healthcare resources [9].

We, and others, have shown that longer-term survival following a critical illness is modulated by non-modifiable patient factors, such as demographics and pre-existing comorbidity, and to a lesser extent by modifiable care process factors, such as intensive care unit (ICU) bed shortages [7,10,11]. Beyond this, there is a growing understanding that the transition to survivorship following discharge from critical care requires effective collaboration with primary care, with recent national guidelines further emphasising the need for holistic management of the persistent and complex health and social care needs that arise during post-critical-care recovery [3,9,12]. Relatively little is known about how critical illness survivors access healthcare following hospital discharge, particularly in outpatient and primary care settings. Furthermore, a more comprehensive understanding of health trajectories and factors determining healthcare resource utilisation (HRU) after critical care is needed to inform service development, including the pivotal role of critical care follow-up clinics, and to serve as a foundation for the future consideration of targeted resource allocation. The population-wide routinely collected healthcare data in Wales, United Kingdom, provide an excellent platform to study this.

Our aims were to compare the usage of primary and secondary care services in the years after a critical care admission with pre-admission HRU; identify which modifiable and non-modifiable factors influence post-critical-care HRU; and investigate whether critical illness survivorship is associated with increased post-hospital HRU compared with that of non-critical-care inpatients.

## 2. Materials and Methods

We conducted a retrospective database cohort study using population-wide routinely collected healthcare data in Wales, United Kingdom.

### 2.1. Data Sources

This study utilised the Wales national trusted research environment (TRE), the Secure Anonymised Information Linkage (SAIL) Databank, a privacy-protecting repository of linked population-scale, individual-level anonymised data sources, including data from NHS Wales [13,14,15,16]. NHS Wales, the main provider of healthcare in Wales, is publicly funded and is free at the point of use, although paid, private healthcare is also available. NHS Wales provides primary care services, including general practice, dentistry, pharmacy, and eye health; secondary care, including elective, urgent, and emergency care; and tertiary care and community health services.

From SAIL, we used the Welsh Critical Care Minimum Dataset (CCDMS) to identify the study cohort; the Welsh Demographic Service Dataset (WDSD) to determine demographics and residence in Wales; the Welsh Longitudinal General Practice (WLGP) covering primary care events; the Emergency Department Dataset (EDDS); the Patient Episode Database for Wales (PEDW) covering inpatient activity; and the Outpatient Dataset (OPD) to calculate health resource utilisation. These data have complete national coverage, except WLGP, which covered about 76% of the population of Wales at the time of data extraction.

### 2.2. Patient Selection

The study cohort included the first recorded admission to adult ICUs in Wales for residents in Wales who were at least 16 years old on the admission date and were discharged alive between 1 April 2006 and 31 December 2017.

Critical care episodes were excluded if patients were discharged to palliative care; transferred to another ICU; lived in Wales for less than 12 months before critical care admission; had missing critical care admission or discharge dates, discharge status, or socioeconomic data; or had low-quality record linkage within SAIL. Only the first admission to critical care was included for each identified patient.

Cohort selection is shown in Figure 1.

### 2.3. Study Variables

The predictor variables of interest included age group, sex, comorbidity level, and level of deprivation. Age was calculated on the date of admission to critical care and was categorised into groups of 16–49 (the reference group), 50–59, 60–69, 70–79, and ≥80 years. The comorbidity level was calculated using a modified Charlson comorbidity index (CCI) calculated on the date of admission to critical care, with a 1-year lookback period, categorised as low (<1), medium (from 1 to less than 10), or high (≥10) [17].

We determined socioeconomic deprivation using the 2011 Welsh Index of Multiple Deprivation (WIMD), the official measure of relative deprivation for small areas in Wales [18]. The overall rank of the 2011 WIMD was calculated from the weighted sum of the following eight deprivation domains: income (23.5%), employment (23.5%), health (14.0%), education (14.0%), geographical access to services (10.0%), housing (5.0%), physical environment (5.0%), and community safety (5.0%). The 2011 WIMD was calculated for each of the 1896 lower layer super output areas (LSOAs) in Wales, which were small geographic area designed by the UK Office for National Statistics (ONS) for the purpose of the 2001 census with an average population of 1500 people. For each patient in our cohort, we used the quintile of the 2011 WIMD overall rank associated with the LSOA of their residential address on the date of admission to critical care.

We also extracted counts of HRU events, including general practitioner (GP) consultations, emergency department (ED) attendances, outpatient attendances, and hospital days, during the 365 days before critical care admission.

The primary outcomes were counts of the aforementioned HRU events during the follow-up period of up to 365 days after discharge from critical care, right-censored by residence in Wales and death. For GP consultations, the follow-up period was further limited by the primary care data coverage [19].

### 2.4. Statistical Analysis

Basic demographic data are presented as counts and percentages. HRU data are presented as medians (with interquartile ranges, IQRs) and means (with standard deviations, SDs).

We fitted negative binomial models using the glm.nb function from the R MASS package, version 7.3-54 (by Brian Ripley et al.), to estimate the associations between the predictor variables and each HRU category utilised by critical illness survivors during the year after discharge. From these models, we present the incidence rate ratios (IRRs) with 95% confidence intervals (CIs).

We used a paired *t*-test to test whether the mean difference between the 1-year pre- and post-critical-care HRU in each HRU category was different from zero. In addition, we used linear models to estimate the associations between the predictor variables and the absolute change in each HRU category between the 12 months before and after critical illness.

To estimate the association of critical care admission and post-hospitalisation HRU, we fitted separate multilevel negative binomial models (using the glmer function from the lme4 package, version 1.1_27.1 by Douglas Bates et al.) for each health resource type, accounting for multiple admissions where these existed. Due to computational limitations, we limited this analysis to patients discharged from critical care in 2015 and a random sample of 200,000 non-critical inpatients discharged from hospitals in the same year.

All models were controlled for the count of corresponding HRU events in the year before critical care.

In all the negative binomial models, the log length of the post-critical-care follow-up out of a maximum of 365 days was used as an offset. Models were based on complete cases only.

Model fit was assessed using quantile–quantile (Q-Q) plots of residuals and using the DHARMa package version 0.4.6 (by Florian Hartig et al.). Appendix A include diagnostic plots for the models.

The analysis was performed in R 4.1.1.

### 2.5. Reporting

The strengthening the reporting of observational studies in epidemiology (STROBE) and the reporting of studies conducted using observational routinely collected health data (RECORD) statements were followed in the reporting of this study [20,21]. The checklists are available in the Appendix A.

## 3. Results

### 3.1. Demographics and Descriptive Analysis

We identified 55,151 patients discharged from critical care in Wales between April 2006 and December 2017 who met the study criteria. The median age was 66.0 years (IQR = 24.3), and 53.7% were male. In total, 37.8% of patients were classed as having high levels of comorbidity, with a median CCI of 7.0 (IQR = 14.0). The level of socioeconomic deprivation was relatively high, with 45.7% of patients occupying the lowest two quintiles of deprivation. Full demographic characteristics are shown in Table 1.

Approximately one out of five (19.2%) patients died within one year of discharge from critical care. The mean follow-up period following discharge was 311.0 days (SD = 116.0) for ED, inpatient, and outpatient data.

Primary care data were available for 75.5% (mean follow-up of 361.9 days, SD = 27.9) before the index critical care admission and for 69.9% (mean follow-up of 328.3 days, SD = 93.5) in the year after discharge.

The mean number of GP consultations per year before critical care was 35.6 (±SD = 23.7), which increased by 11.0% to 39.5 (±26.5). The mean number of outpatient attendances increased by 17.3% from 5.2 (±6.1) to 6.1 (±7.1). The mean length of stay in hospital increased by 86.5% from 5.2 (±22.1) to 9.7 (±22.1) days. Conversely, the mean number of ED attendances in the year following a critical illness decreased by 30% from 1.2 (±1.8) to 0.8 (±1.8). The *p* values for all paired *t*-tests were <2 × 10^−16^.

### 3.2. Factors Associated with Post-Critical-Care HRU

Compared to younger patients, critical care survivors over 50 years of age demonstrated greater HRU in the year following critical illness across most healthcare levels. Those over 50 years of age were shown to have 11–15% more GP consultations, 47–196% more days spent in hospital, and 3–13% more outpatient visits, except those over 80 years old who had at least 15% fewer outpatient visits compared to the younger age groups (Table 2). Interestingly, all age groups over 50 demonstrated 17–23% fewer ED attendances. Age groups over 50 were also associated with increases of between 30 and 77 in the number of days spent in hospital following critical care compared with the previous year, compared with an increase of 17.7 days among people younger than 50 (Table 3).

High comorbidity prior to critical care admission was also associated with significantly increased HRU following critical care, including 22% more GP consultations (incidence rate ratio (IRR) = 1.22 [1.20, 1.24]), 43% more ED attendances (IRR: 1.43 [1.36, 1.50]), 62% more days in hospital (IRR: 1.62 [1.57, 1.67]), and 47% more outpatient attendances (IRR: 1.47 [1.44, 1.50]) compared to no or low levels of comorbidity. High comorbidity was associated with an average increase of 45.3 in the number of days spent in hospital following critical care, compared to 17.7 for no or low comorbidity (average difference: 27.62; 95% CI: 25.50, 29.74; Table 3).

Compared with patients from the least deprived areas, those from the most deprived areas demonstrated greater levels of seeking emergency care in the year following critical care, with 24% more ED attendances (IRR: 1.24 [1.16, 1.32]) and 13% longer hospital stays (IRR: 1.13 [1.09, 1.17]). The increase in the number of days in hospital after clinical illness was on average 6.94 (4.35, 9.53) days longer in the most deprived group compared with the least deprived group (Table 3)

Sex was not a clinically significant predictor of post-critical-care HRU, although females used slightly less emergency and secondary care than males (Table 2 and Table 3).

### 3.3. Comparison with the Wider Hospital Population

We compared 5204 critical care survivors discharged in 2015 with a sample of 200,000 from 831,558 non-critical-care inpatients discharged in the same year.

The characteristics of both groups are shown in Table 4. Critical care survivors were older, with a median age of 66.0 years (IQR: 23.7), compared to 61.5 (32.8) in the rest of the hospital population. They showed slightly higher levels of deprivation and higher proportions of males than the non-critical-care inpatients (54.5% vs. 42.6%). Both groups had similar follow-up periods.

Critical illness was associated with 7% more GP consultations (IRR = 1.07 [1.06, 1.08]), 12% more ED attendances (1.12 [1.07, 1.16]), 9% more outpatient attendances (1.09 [1.07, 1.11]), and 68% longer hospital stays (1.68 [1.63, 1.74]) during the year following hospital discharge (Table 5).

## 4. Discussion

In a large database of critical care survivors in Wales, over a decade, we found that patients demonstrated significantly higher levels of HRU following discharge from critical care compared to baseline levels prior to their critical illness. This increased level of HRU was observed for both primary and secondary care services. In addition, critical care survivors demonstrated higher levels of HRU compared to non-critical inpatients, with 68% longer hospital stays for readmissions and 7–12% more GP consultations, outpatient attendances, and ED attendances.

Our findings extend prior work in this area that demonstrated increasing rates of HRU in critical care survivors by adding further data regarding changes in the patterns of HRU and providing a more comprehensive understanding of subsequent HRU in the primary care setting [5,11,22].

Several studies have identified that critical care survivors are at a greater risk of hospital readmission in both the short and long term following discharge from critical care, along with higher rates of mortality compared to non-critical-care patients [6,10,23]. In 2016, Hill et al. found that almost a third of critical care discharges were hospitalised within the first 6 months [5]. Hirshberg et al. found similar rates, with 26% readmitted within 90 days of discharge and 43% readmitted within 1 year following discharge [24]. Our findings that both GP and outpatient appointments were significantly increased provide further context to interpret the increasing number of hospital readmissions and higher mortality rates that are observed. GPs play a crucial role in managing chronic conditions and signposting the need for more specialised services. Although GP and outpatient attendances increased significantly after critical-care discharge and were more frequent compared to the non-critical-care group, our results do not suggest an over two-fold increase in primary care HRU, as demonstrated in a previous Dutch study [25]. The notification of GPs about a patient’s critical care stay is not a standardised and automated process in Wales, which could explain some of the differences, as it is not a standard practice for the primary care provider to actively seek contact. Conversely, patient behaviour and a lack of understanding of the primary and secondary care arrangements following a critical care stay could further reduce primary care contacts.

We demonstrated that previously identified predictors of long-term mortality in the same cohort also influence higher levels of HRU, with increasing age and multiple comorbidities being the most significant predictors of increased HRU in the year following critical illness [10].

Our finding that older age was a strong predictor of higher HRU following critical illness is consistent with other studies [5,26,27]. Age is a commonly used surrogate marker for frailty and functional measures that are harder to account for in population-level studies, which likely underpin the significantly greater lengths of stay following admission that were observed in this study and the higher levels of outpatient HRU. Interestingly, we found that although those aged over 50 years demonstrated significantly higher numbers of GP consultations and outpatient visits, they also showed 20–26% fewer ED attendances. Of note, HRU categories are potentially partially interdependent. The fewer post-critical-care ED attendances observed in the more elderly population may be offset by the increased use of other modalities of care, including longer periods of hospitalisation. In addition, this could also be due to perceived or actual limited access to emergency departments for more elderly patients [28].

A greater number of comorbidities was associated with a significantly higher level of HRU in the outpatient setting, with patients with higher levels of comorbidity demonstrating 21% more GP attendances and 48% more outpatient attendances than in the prior year. This potentially suggests a worsening of existing health conditions following a critical illness or a decrease in a patient’s ability to manage their pre-existing conditions, leading to a need to seek further support and a likely coexistent further deterioration in general levels of health [11,29]. It is possible that this higher HRU in patients with higher levels of comorbidity may be due in part to a worsening of comorbid conditions prior to and contributing to the initial critical illness. Similar to our findings, other studies have shown a persistent and stable increase in HRU over subsequent years following a critical illness, suggesting an increasing burden of comorbid conditions after discharge [11]. In addition, a study by Hill et al. identified that 28% of all readmissions following critical illnesses were for diagnoses related to the initial index admission, suggesting that the worsening of pre-existing illnesses following discharge likely accounts for a substantial portion of readmissions [5]. Jouan et al. also observed an increased frequency of major comorbidities related to renal, respiratory, and cardiac functions in the post-ICU period compared to the pre-ICU period, which further supports this view [11]. Our results further highlight the need for better communication between secondary and primary care services during the discharge process following critical care.

Several studies have identified a relationship between levels of deprivation and ICU outcomes [30]. Garland et al. identified a steady gradient of declining hospital mortality with rising income [22]. Welch et al. highlighted a strong association between an increasing level of deprivation and both increased incidence of admission to ICU and increased mortality following admission [31]. We have also shown previously that an increased level of deprivation is an independent factor of long-term outcomes in critical care survivors [10]. With regard to post-critical-care activity, we found that patients from the most deprived areas had 24% more ED attendances than those from the least deprived areas. Although increased HRU has been reported in more deprived areas in the US, to our knowledge our results are the first to quantify this difference in a post-critical-care setting [24]. In the UK, higher deprivation is not thought to be a financial barrier to ED attendance, unlike in countries with private or hybrid payer models.

Our description of factors predicting increased post-critical-care HRU are of particular interest in the current climate, with increasing numbers of critical care admissions due to the COVID-19 pandemic leading to an increasing population of critical care survivors with complex health and social needs. Further studies will be required to further characterise the trajectory of healthcare use for COVID-19-specific critical care survivors; however, it is likely the findings discussed in this paper will remain applicable.

There are several limitations to our study. We have not investigated whether our outcomes are affected by the type or duration of organ support during admission. Notably, previous reports suggested this has a minimal impact on 1-year HRU [32]. In addition, our analysis did not include information on illness severity at the time of ICU admission.

Although age is most commonly used in studies of this nature and, together with CCI, it can be used as a surrogate, further data on frailty scores and pre- and post-hospital levels of functional status may also provide better insights into particular patient subgroups with more complex post-critical-care recovery [33,34].

In addition, we were unable to collect data regarding calls to 999 and the use of emergency ambulances, which would have provided more context for the interpretation of the reported ED attendances.

Finally, although the model fit was generally good, there were variable degrees of skewness towards the highest event counts. Therefore, our estimates should be interpreted with caution in patients with the highest HRU.

## 5. Conclusions

Our findings highlight population-level consequences of critical care survivorship, identifying key patterns of HRU and the factors that best predict increased post-critical-illness HRU. We identified that critical care survivors use significantly greater amounts of healthcare resources in primary and secondary care settings compared to both their prior level of HRU before admission and that of non-critical-care inpatients. We also demonstrated that previously identified predictors of mortality in critical care survivors also predict increases in HRU following critical care discharge.

These findings could be used to design more effective interventions aiming to ensure key subgroups of critical care survivors are identified and adequately supported. Further research is required to identify a wider set of factors, especially modifiable factors, that lead to the increased HRU among survivors in order to better inform resource allocation. Future trials should aim to address the burden critical care survivorship places on primary and secondary care.

## Figures and Tables

**Figure 1 jcm-12-00872-f001:**
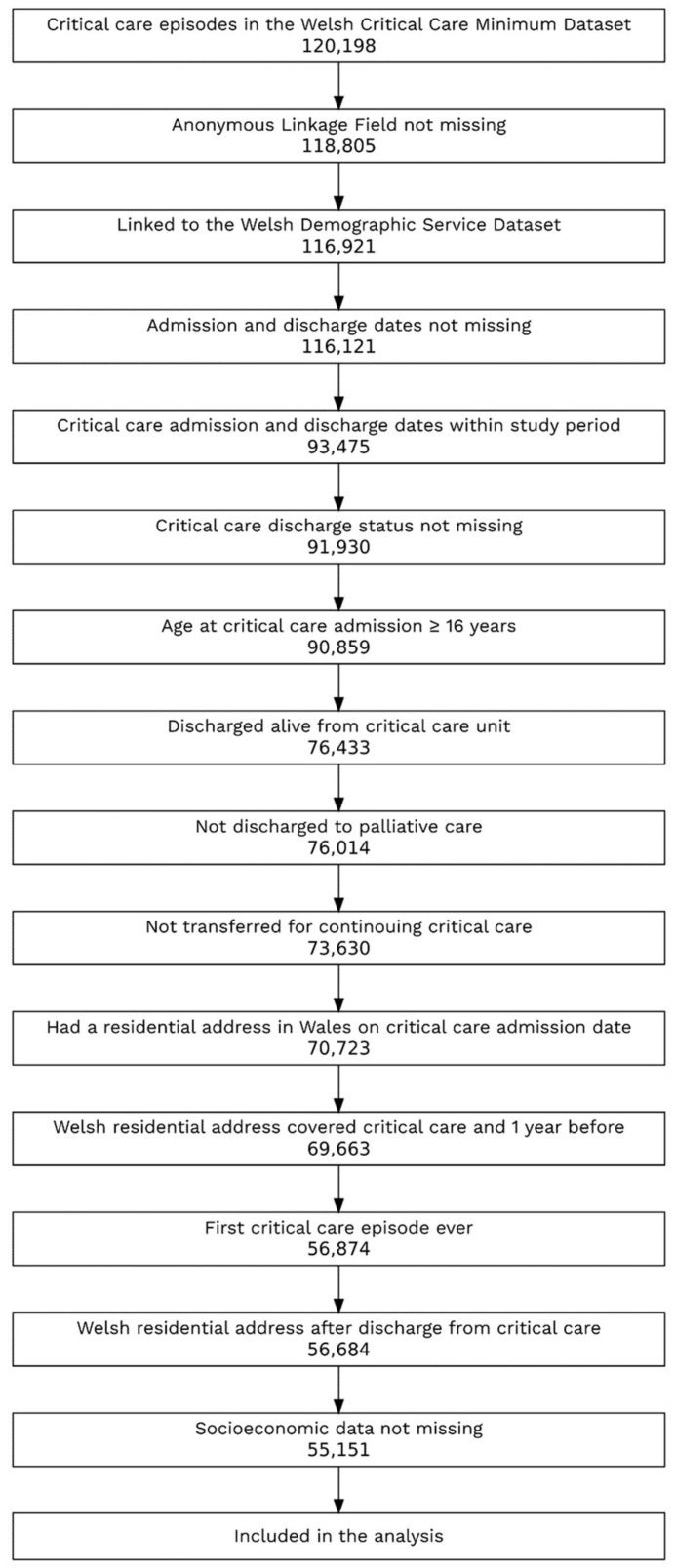
Cohort selection.

**Table 1 jcm-12-00872-t001:** Characteristics of the critical care cohort.

	Overall
	(*n* = 55,151)
Sex	
Male	29,604 (53.7%)
Female	25,547 (46.3%)
Age	
Median (IQR)	66.0 (24.3)
Mean (SD)	62.4 (17.9)
Groups	
16–49 years	12,624 (22.9%)
50–59 years	7752 (14.1%)
60–69 years	12,716 (23.1%)
70–79 years	13,651 (24.8%)
80+ years	8408 (15.2%)
2011 Welsh Index of Multiple Deprivation, quintiles	
Most deprived	12,964 (23.5%)
Next most deprived	12,224 (22.2%)
Middle deprivation	11,758 (21.3%)
Next least deprived	9549 (17.3%)
Least deprived	8656 (15.7%)
Charlson Comorbidity Index (modified)	
Median (IQR)	7.00 (14.0)
Mean (SD)	9.18 (9.70)
Categories	
Low (<1)	16,496 (29.9%)
Medium (1 to <10)	17,793 (32.3%)
High (≥10)	20,862 (37.8%)
Follow-up in SAIL	
1 year before critical care	
Median (IQR)	365 (0)
Mean (SD)	365 (0)
1 year after critical care	
Median (IQR)	365 (0)
Mean (SD)	311 (116)
Follow-up in the GP data	
1 year before critical care	
Median (IQR)	365 (0)
Mean (SD)	362 (27.9)
Missing	13,534 (24.5%)
1 year after critical care	
Median (IQR)	365 (0)
Mean (SD)	328 (93.5)
Missing	16,607 (30.1%)
GP events	
1 year before critical care	
Median (IQR)	32.0 (28.0)
Mean (SD)	35.6 (23.7)
Missing	13,534 (24.5%)
1 year after critical care	
Median (IQR)	37.0 (34.0)
Mean (SD)	39.5 (26.5)
Missing	16,607 (30.1%)
Outpatient attendances	
1 year before critical care	
Median (IQR)	3.00 (6.00)
Mean (SD)	5.17 (6.13)
1 year after critical care	
Median (IQR)	4.00 (7.56)
Mean (SD)	6.11 (7.06)
ED attendances	
1 year before critical care	
Median (IQR)	1.00 (2.00)
Mean (SD)	1.21 (1.78)
1 year after critical care	
Median (IQR)	0.00 (1.00)
Mean (SD)	0.85 (1.79)
Length of hospital stay	
1 year before critical care	
Median (IQR)	2.00 (10.0)
Mean (SD)	9.67 (22.1)
1 year after critical care	
Median (IQR)	10.0 (26.0)
Mean (SD)	26.2 (44.0)

ED: emergency department; GP: general practitioner; IQR: interquartile range; SAIL: Secure Anonymised Information Linkage; SD: standard deviation.

**Table 2 jcm-12-00872-t002:** Factors associated with 1-year post-critical-care HRU.

	GP Consultations		Outpatient Attendances		ED Attendances		Length of Hospital Stay	
	IRR (95% CI)	*p* Value	IRR (95% CI)	*p* Value	IRR (95% CI)	*p* Value	IRR (95% CI)	*p* Value
Most deprived	0.983 (0.967, 0.999)	0.043	0.967 (0.944, 0.991)	0.007	1.237 (1.158, 1.320)	<0.001	1.128 (1.085, 1.171)	<0.001
Next most deprived	0.997 (0.981, 1.014)	0.759	0.982 (0.959, 1.006)	0.149	1.157 (1.083, 1.236)	0.000	1.061 (1.021, 1.102)	0.003
Middle deprivation	1.018 (1.001, 1.035)	0.040	0.946 (0.923, 0.969)	<0.001	1.104 (1.033, 1.179)	0.004	1.049 (1.009, 1.091)	0.015
Next least deprived	1.014 (0.996, 1.032)	0.127	0.944 (0.920, 0.968)	<0.001	1.107 (1.033, 1.187)	0.004	0.995 (0.956, 1.037)	0.827
CCI: medium	1.133 (1.118, 1.148)	<0.001	1.224 (1.200, 1.248)	<0.001	1.200 (1.141, 1.263)	<0.001	1.003 (0.973, 1.035)	0.825
CCI: high	1.221 (1.204, 1.238)	<0.001	1.474 (1.445, 1.504)	<0.001	1.429 (1.357, 1.505)	<0.001	1.622 (1.572, 1.674)	<0.001
Age 50–59	1.141 (1.122, 1.161)	<0.001	1.123 (1.095, 1.152)	<0.001	0.798 (0.748, 0.851)	<0.001	1.470 (1.413, 1.531)	<0.001
Age 60–69	1.150 (1.132, 1.168)	<0.001	1.127 (1.102, 1.153)	<0.001	0.783 (0.738, 0.830)	<0.001	1.655 (1.596, 1.716)	<0.001
Age 70–79	1.134 (1.116, 1.152)	<0.001	1.029 (1.006, 1.053)	0.013	0.768 (0.723, 0.815)	<0.001	2.059 (1.986, 2.135)	<0.001
Age 80+	1.114 (1.093, 1.135)	<0.001	0.853 (0.831, 0.877)	<0.001	0.831 (0.777, 0.890)	<0.001	2.963 (2.844, 3.086)	<0.001
Female	1.005 (0.995, 1.015)	0.340	0.979 (0.964, 0.994)	0.005	0.964 (0.928, 1.002)	0.065	0.972 (0.949, 0.995)	0.017
Event count in the previous year	1.013 (1.013, 1.014)	<0.001	1.058 (1.056, 1.059)	<0.001	1.258 (1.246, 1.269)	<0.001	1.013 (1.013, 1.014)	<0.001
Intercept	0.062 (0.060, 0.063)	<0.001	0.012 (0.011, 0.012)	<0.001	0.002 (0.002, 0.002)	<0.001	0.060 (0.057, 0.062)	<0.001

CI: confidence interval; CCI: Charlson comorbidity index; ED: emergency department; GP: general practitioner; IRR: incidence rate ratio.

**Table 3 jcm-12-00872-t003:** Factors associated with the difference between 1-year pre-critical-care HRU and 1-year post-critical-care HRU.

	GP Consultations		Outpatient Attendances		ED Attendances		Length of Hospital Stay	
	Change (95% CI)	*p* Value	Change (95% CI)	*p* Value	Change (95% CI)	*p* Value	Change (95% CI)	*p* Value
Most deprived	−0.47 (−1.49, 0.54)	0.360	−0.08 (−0.35, 0.20)	0.580	0.24 (0.12, 0.36)	<0.001	6.94 (4.35, 9.53)	<0.001
Next most deprived	0.25 (−0.78, 1.28)	0.633	0.03 (−0.25, 0.30)	0.840	0.18 (0.06, 0.31)	0.003	3.61 (1.01, 6.22)	0.007
Middle deprivation	1.42 (0.38, 2.46)	0.007	−0.22 (−0.50, 0.05)	0.111	0.10 (−0.02, 0.22)	0.099	3.45 (0.83, 6.07)	0.010
Next least deprived	0.69 (−0.42, 1.80)	0.222	−0.30 (−0.58, −0.01)	0.045	0.18 (0.05, 0.30)	0.007	−0.18 (−2.92, 2.57)	0.900
CCI: medium	3.95 (3.12, 4.77)	<0.001	1.10 (0.88, 1.32)	<0.001	0.20 (0.11, 0.30)	<0.001	−0.76 (−2.83, 1.31)	0.473
CCI: high	8.83 (7.97, 9.69)	<0.001	2.49 (2.27, 2.72)	<0.001	0.29 (0.19, 0.39)	<0.001	27.62 (25.50, 29.74)	<0.001
Age 50–59	4.03 (2.96, 5.09)	<0.001	0.68 (0.40, 0.97)	<0.001	−0.21 (−0.34, −0.09)	<0.001	12.49 (9.77, 15.21)	<0.001
Age 60–69	3.64 (2.67, 4.61)	<0.001	0.64 (0.38, 0.90)	<0.001	−0.21 (−0.33, −0.10)	<0.001	17.96 (15.51, 20.40)	<0.001
Age 70–79	3.23 (2.25, 4.21)	<0.001	−0.28 (−0.54, −0.02)	0.032	−0.22 (−0.34, −0.11)	<0.001	31.12 (28.68, 33.57)	<0.001
Age 80+	2.98 (1.85, 4.12)	<0.001	−1.58 (−1.87, −1.29)	<0.001	−0.24 (−0.37, −0.12)	<0.001	59.13 (56.37, 61.89)	<0.001
Female	−0.37 (−1.01, 0.26)	0.252	−0.26 (−0.43, −0.10)	0.002	−0.05 (−0.12, 0.03)	0.198	0.13 (−1.46, 1.72)	0.874
Event count in the previous year	−0.31 (−0.32, −0.30)	<0.001	−0.45 (−0.47, −0.44)	<0.001	−0.52 (−0.54, −0.50)	<0.001	−0.13 (−0.16, −0.09)	<0.001
Intercept	16.65 (15.55, 17.75)	<0.001	3.74 (3.45, 4.04)	<0.001	0.48 (0.35, 0.62)	<0.001	18.92 (16.18, 21.66)	<0.001

CI: confidence interval; CCI: Charlson comorbidity index; ED: emergency department; GP: general practitioner.

**Table 4 jcm-12-00872-t004:** Characteristics of the critical care cohort and the non-critical inpatient cohort in 2015.

	Critical Care Sub-Cohort	Non-Critical-Care Hospital Population	Overall
	(*n* = 5204)	(*n* = 200,000)	(*n* = 205,204)
Sex			
Male	2837 (54.5%)	85,109 (42.6%)	87,946 (42.9%)
Female	2367 (45.5%)	114,891 (57.4%)	117,258 (57.1%)
Age			
Median (IQR)	66.0 (23.7)	61.5 (32.8)	61.7 (32.5)
Mean (SD)	62.5 (17.7)	57.9 (20.4)	58.0 (20.4)
Groups			
<50	1156 (22.2%)	67,884 (33.9%)	69,040 (33.6%)
50–59	797 (15.3%)	27,417 (13.7%)	28,214 (13.7%)
60–69	1227 (23.6%)	37,888 (18.9%)	39,115 (19.1%)
70–79	1239 (23.8%)	38,249 (19.1%)	39,488 (19.2%)
80+	785 (15.1%)	28,562 (14.3%)	29,347 (14.3%)
2011 Welsh Index of Multiple Deprivation, quintiles			
Most deprived	1164 (22.4%)	41,722 (20.9%)	42,886 (20.9%)
Next most deprived	1162 (22.3%)	42,030 (21.0%)	43,192 (21.0%)
Middle deprivation	1161 (22.3%)	40,819 (20.4%)	41,980 (20.5%)
Next least deprived	975 (18.7%)	38,332 (19.2%)	39,307 (19.2%)
Least deprived	742 (14.3%)	37,097 (18.5%)	37,839 (18.4%)
Charlson Comorbidity Index (modified)			
Median (IQR)	7.00 (14.0)	2.00 (9.00)	3.00 (10.0)
Mean (SD)	9.07 (9.42)	6.35 (8.99)	6.42 (9.01)
Categories			
Low (<1)	1505 (28.9%)	99,752 (49.9%)	101,257 (49.3%)
Medium (1 to <10)	1740 (33.4%)	50,630 (25.3%)	52,370 (25.5%)
High (≥10)	1959 (37.6%)	49,618 (24.8%)	51,577 (25.1%)
Follow-up in SAIL			
1 year before critical care			
Median (IQR)	365 (0)	365 (0)	365 (0)
Mean (SD)	365 (0)	361 (31.5)	361 (31.1)
1 year after critical care			
Median (IQR)	365 (0)	365 (0)	365 (0)
Mean (SD)	330 (90.9)	339 (78.7)	338 (79.1)
Follow-up in the GP data			
1 year before critical care			
Median (IQR)	365 (0)	365 (0)	365 (0)
Mean (SD)	361 (30.8)	358 (40.3)	358 (40.1)
Missing	1275 (24.5%)	47,604 (23.8%)	48,879 (23.8%)
1 year after critical care			
Median (IQR)	365 (0)	365 (0)	365 (0)
Mean (SD)	327 (95.2)	335 (83.2)	335 (83.6)
Missing	1276 (24.5%)	47,547 (23.8%)	48,823 (23.8%)
GP consultations			
1 year before critical care			
Median (IQR)	26.0 (43.0)	23.0 (36.0)	24.0 (36.0)
Mean (SD)	28.0 (25.8)	26.9 (24.9)	27.0 (24.9)
1 year after critical care			
Median (IQR)	31.0 (52.0)	22.0 (39.0)	22.0 (40.0)
Mean (SD)	32.6 (29.4)	26.0 (25.9)	26.2 (26.0)
ED attendances			
1 year before critical care			
Median (IQR)	1.00 (2.00)	0 (1.00)	0 (1.00)
Mean (SD)	1.20 (1.77)	0.861 (1.81)	0.870 (1.81)
1 year after critical care			
Median (IQR)	0 (1.00)	0 (1.00)	0 (1.00)
Mean (SD)	0.960 (1.67)	0.756 (1.92)	0.761 (1.91)
Outpatient attendances			
1 year before critical care			
Median (IQR)	3.00 (6.00)	4.00 (7.00)	4.00 (7.00)
Mean (SD)	5.14 (6.08)	5.76 (6.60)	5.74 (6.59)
1 year after critical care			
Median (IQR)	5.00 (7.00)	4.00 (7.00)	4.00 (7.00)
Mean (SD)	6.74 (6.92)	5.65 (6.89)	5.68 (6.89)
Length of hospital stay, days			
1 year before critical care			
Median (IQR)	1.00 (7.00)	2.00 (12.0)	2.00 (12.0)
Mean (SD)	7.83 (17.8)	12.9 (29.6)	12.8 (29.3)
1 year after critical care			
Median (IQR)	12.0 (29.0)	3.00 (14.0)	3.00 (15.0)
Mean (SD)	29.0 (44.9)	15.0 (32.7)	15.3 (33.1)

ED: emergency department; GP: general practitioner; IQR: interquartile range; SAIL: Secure Anonymised Information Linkage; SD: standard deviation.

**Table 5 jcm-12-00872-t005:** Associations between critical care admission and health resource utilisation within 1 year of discharge from hospital.

	GP Consultations		Outpatient Attendances		ED Attendances		Length of Hospital Stay	
	IRR (95% CI)	*p* Value	IRR (95% CI)	*p* Value	IRR (95% CI)	*p* Value	IRR (95% CI)	*p* Value
Critical care	1.07 (1.06, 1.08)	0.001	1.09 (1.07, 1.11)	<0.001	1.12 (1.07, 1.16)	<0.001	1.68 (1.63, 1.74)	<0.001
CCI: medium	1.21 (1.20, 1.22)	<0.001	1.41 (1.40, 1.43)	<0.001	1.35 (1.32, 1.38)	<0.001	1.70 (1.66, 1.74)	<0.001
CCI: high	1.21 (1.21, 1.22)	<0.001	1.56 (1.54, 1.58)	<0.001	1.75 (1.70, 1.79)	<0.001	2.65 (2.59, 2.72)	<0.001
Most deprived	1.10 (1.09, 1.11)	<0.001	1.00 (0.98, 1.02)	0.870	1.47 (1.42, 1.52)	<0.001	1.29 (1.24, 1.34)	<0.001
Next most deprived	1.08 (1.07, 1.10)	<0.001	0.99 (0.97, 1.01)	0.422	1.35 (1.31, 1.40)	<0.001	1.19 (1.14, 1.23)	<0.001
Middle deprivation	1.07 (1.06, 1.08)	<0.001	0.98 (0.97, 1.00)	0.066	1.27 (1.23, 1.31)	<0.001	1.12 (1.08, 1.17)	<0.001
Next least deprived	1.06 (1.05, 1.07)	<0.001	0.99 (0.97, 1.01)	0.372	1.09 (1.05, 1.12)	<0.001	1.04 (1.00, 1.08)	<0.060
Age 50–59	1.39 (1.37, 1.40)	<0.001	1.36 (1.33, 1.38)	<0.001	0.86 (0.83, 0.89)	<0.001	1.13 (1.09, 1.18)	<0.001
Age 60–69	1.58 (1.57, 1.60)	<0.001	1.50 (1.48, 1.53)	<0.001	0.84 (0.81, 0.86)	<0.001	1.55 (1.49, 1.61)	<0.001
Age 70–79	1.74 (1.72, 1.75)	<0.001	1.55 (1.52, 1.57)	<0.001	1.00 (0.97, 1.03)	0.862	2.16 (2.08, 2.24)	<0.001
Age 80+	1.87 (1.85, 1.89)	<0.001	1.24 (1.21, 1.26)	<0.001	1.52 (1.47, 1.57)	<0.001	3.87 (3.72, 4.03)	<0.001
Female	1.04 (1.03, 1.04)	<0.001	0.95 (0.94, 0.96)	<0.001	0.90 (0.88, 0.92)	<0.001	1.09 (1.06, 1.12)	<0.001
Event count in the previous year	1.01 (1.01, 1.01)	<0.001	1.01 (1.01, 1.01)	<0.001	1.08 (1.08, 1.08)	<0.001	0.99 (0.99, 0.99)	<0.001
Intercept	0.04 (0.03, 0.04)	<0.001	0.01 (0.01, 0.01)	<0.001	0.00 (0.00, 0.00)	<0.001	0.00 (0.00, 0.00)	<0.001

CI: confidence interval; CCI: Charlson comorbidity index; ED: emergency department; GP: general practitioner; IRR: incidence rate ratio.

## Data Availability

The anonymised person-level data supporting the conclusions of this article are held by the SAIL Databank (https://saildatabank.com/ [accessed on 20 January 2023]) and are restricted and not publicly available but can be accessed upon reasonable request, with permission from SAIL. All proposals to use SAIL are carefully reviewed by an independent information governance review panel (IGRP) that includes members of the public to ensure the proper and appropriate use of data (https://www.saildatabank.com/application-process [accessed on 20 January 2023]). When approved, access is then provided through the SAIL Gateway, a privacy-protecting safe haven and a secure remote access system.

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
