# Peer review of "Patterns of Healthcare Resource Utilisation of Critical Care Survivors between 2006 and 2017 in Wales: A Population-Based Study"

_jcm, 2023, doi:10.3390/jcm12030872_

Round 1
Reviewer 1 Report
This study investigated healthcare resource utilization among ICU survivors. I think the strength of this study is that it analyzed the data from all patients in Wales. This paper is well written and worth reporting. However, there are some major concerns:
C1. Increased utilization of healthcare resource by ICU survivors is well-known. How is this study different from previous studies?
C2. Why did ED attendance decrease in some groups as opposed to other healthcare resource utilization?
C3. Data on ICU were not presented in this paper. What diseases were the patients admitted to the ICU for? How were the patients treated in the ICU (such as MV, CRRT)?
C4. There is an incorrect wording.
Table 1- ED attendances - 1 year before critical care -> 1 year after critical care
Author Response
We would like to thank the reviewer for the thoughtful comments and efforts towards improving our manuscript. Please find below our point-by-point responses.
C1. Increased utilization of healthcare resource by ICU survivors is well-known. How is this study different from previous studies?
Our response: To our knowledge, there is no detailed contemporary data this from the United Kingdom. We believe that our findings represent important baseline for studies that aim to influence health source utilisation among critical care survivors. In addition, we present interesting data regarding higher emergency department attendance among the younger survivors, as pointed out by the reviewer.
C2. Why did ED attendance decrease in some groups as opposed to other healthcare resource utilization?
Our response: The pattern of lower emergency department attendance among the older survivors (over 50 years old) is an interesting and novel observation. However, it will require further analysis to understand the potential drivers and the possible role of competing events (e.g., other categories of health resource utilisation), which we did not consider in our current work. We anticipate that this would require a mixed methods approach as population level healthcare data is unlikely to capture all the drivers behind this finding.
C3. Data on ICU were not presented in this paper. What diseases were the patients admitted to the ICU for? How were the patients treated in the ICU (such as MV, CRRT)?
Our response: Detailed clinical data about ICU admissions were available in the ICNARC Case Mix Programme Dataset (https://www.icnarc.org/Our-Audit/Audits/Cmp/About). However, less than half of people in our cohort could be matched to the dataset that we have access to. Therefore, for this analysis, we decided to use representative population-wide data instead of clinically detailed data that is non-representative. We are developing a more robust methodology which will enable us to map the different datasets in the SAIL Databank more closely. Also, we are in the process of incorporating ICNARC data for a much longer look-back period, however, this takes considerable amount of time. We hope that in the future we will be able to provide more detailed ICU level data, without losing the power of the national population-level datasets.
C4. There is an incorrect wording.
Table 1- ED attendances - 1 year before critical care -> 1 year after critical care
Our response: We apologise for this typo, which we have corrected in the revised manuscript.
Reviewer 2 Report
Thank you very much for inviting me to review this interesting manuscript identifying key patterns of healthcare resource utilization (HRU) by critical care survivors. A better understanding of the role of the factors that predict increased post-critical illness HRU may help design more effective support for this group of patients.
The manuscript is well written, clear and easy to read. The title properly reflect the subject of the paper, the methods used are appropriate, and the data supports the conclusions.
Minor Issues
1) Introduction: 49-51 lines. I suggest identifying which non-modifiable and modifiable factors are associated with long-term survival after critical illness.
2) Materials and Methods. 101-103 lines. Provide the levels of deprivation (indicated in Table 1) and explain how each level differs, as the reader may have difficulty understanding the concept.
3) It would be useful to clarify what forms of healthcare are available in Wales (public or privately funded healthcare?) and what the primary and secondary care services included.
4) The meaning of the abbreviation IRR should be explained in the Method or Result section (below Table 2).
Author Response
We would like to thank the reviewer for the thoughtful comments and efforts towards improving our manuscript. Please find below our point-by-point responses.
Point 1: Introduction: 49-51 lines. I suggest identifying which non-modifiable and modifiable factors are associated with long-term survival after critical illness.
Our response: We have added examples of non-modifiable and modifiable factors that have been found to be associated with long-term survival after critical illness:
We, and others, have shown that longer-term survival following a critical illness is modulated by non-modifiable patient factors, such as demographics and pre-existing comorbidity, and to a lesser extent by modifiable care process factors, such as intensive care unit (ICU) bed shortage [7, 10, 11].
Point 2: Materials and Methods. 101-103 lines. Provide the levels of deprivation (indicated in Table 1) and explain how each level differs, as the reader may have difficulty understanding the concept.
Our response: We have added the following details about the 2011 Welsh Index of Multiple Deprivation, including the fact it is a relative measure of deprivation between small areas, it is composed of eight weighted deprivation domains, and that, for each patient in our cohort, we used the quintile of the overall rank assocaited with their address:
We determined socioeconomic deprivation using the 2011 Welsh Index of Multiple Deprivation (WIMD), the official measure of relative deprivation for small areas in Wales. The overall rank of the WIMD 2011 was calculated from the weighted sum of the following eight deprivation domains: income (23.5%), employment (23.5%), health (14.0%), education (14.0%), geographical access to services (10.0%), housing (5.0%), physical environment (5.0%), and community safety (5.0%). The WIMD 2011 was calculated for each of the 1,896 Lower Layer Super Output Areas (LSOAs) in Wales, which were small area geography designed by the UK Office for National Statistics (ONS) for the purpose of the 2001 census with an average population of 1,500 people. For each patient in our cohort, we used the quintile of WIMD 2011 overall rank associated with the LSOA of their residential address at the date of admission to critical care.
Point 3: It would be useful to clarify what forms of healthcare are available in Wales (public or privately funded healthcare?) and what the primary and secondary care services included.
Our response: We have added the following summary about healthcare in Wales:
This study utilised the Wales national trusted research environment (TRE), the Secure Anonymised Information Linkage (SAIL) Databank, a privacy-protecting repository of linked population-scale, individual-level anonymised data sources, including data from NHS Wales [13–16]. NHS Wales is the main provider of healthcare in Wales, is publicly funded, and is free at the point of use, although paid, private healthcare is also available. NHS Wales provides primary care services, including general practice, dentistry, pharmacy, and eye health; secondary care, including elective, urgent, and emergency care; and tertiary care and community health services.
Point 4: The meaning of the abbreviation IRR should be explained in the Method or Result section (below Table 2).
Our response: We have now defined this abbreviation in the Methods section and in the table footers, were appropriate:
We fitted negative binomial models ... From these models, we present the incidence rate ratio (IRR) with the 95% confidence interval (CI).
Round 2
Reviewer 1 Report
This paper has been appropriately revised.